# “Green” nZVI-Biochar as Fenton Catalyst: Perspective of Closing-the-Loop in Wastewater Treatment

**DOI:** 10.3390/molecules28031425

**Published:** 2023-02-02

**Authors:** Anita Leovac Maćerak, Aleksandra Kulić Mandić, Vesna Pešić, Dragana Tomašević Pilipović, Milena Bečelić-Tomin, Djurdja Kerkez

**Affiliations:** Department of Chemistry, Biochemistry and Environmental Protection, Faculty of Sciences, University of Novi Sad, Trg Dositeja Obradovića 3, 21000 Novi Sad, Serbia

**Keywords:** nano zero-valent iron, biochar, sewage sludge, catalyst, nutrient release, circular economy

## Abstract

In the framework of wastewater treatment plants, sewage sludge can be directed to biochar production, which when coupled with an external iron source has the potential to be used as a carbon–iron composite material for treating various organic pollutants in advanced oxidation processes. In this research, “green” synthesized nano zero-valent iron (nZVI) supported on sewage sludge-based biochar (BC)–nZVI-BC was used in the Fenton process for the degradation of the recalcitrant organic molecule. In this way, the circular economy principles were supported within wastewater treatment with immediate loop closing; unlike previous papers, where only the water treatment was assessed, the authors proposed a new approach to wastewater treatment, combining solutions for both water and sludge. The following phases were implemented: synthesis and characterization of nano zero-valent iron supported on sewage sludge-based biochar (nZVI-BC); optimization of organic pollutant removal (Reactive Blue 4 as the model pollutant) by nZVI-BC in the Fenton process, using a Definitive Screening Design (DSD) model; reuse of the obtained Fenton sludge, as an additional catalytic material, under previously optimized conditions; and assessment of the exhausted Fenton sludge’s ability to be used as a source of nutrients. nZVI-BC was used in the Fenton treatment for the degradation of Reactive Blue 4—a model substance containing a complex and stable anthraquinone structure. The DSD model proposes a high dye-removal efficiency of 95.02% under the following optimal conditions: [RB4] = 50 mg/L, [nZVI] = 200 mg/L, [H_2_O_2_] = 10 mM. pH correction was not performed (pH = 3.2). Afterwards, the remaining Fenton sludge, which was thermally treated (named FS_treated_), was applied as a heterogeneous catalyst under the same optimal conditions with a near-complete organic molecule degradation (99.56% ± 0.15). It could be clearly noticed that the cumulative amount of released nutrients significantly increased with the number of leaching experiments. The highest cumulative amounts of released K, Ca, Mg, Na, and P were therefore observed at the fifth leaching cycle (6.40, 1.66, 1.12, 0.62, 0.48 and 58.2 mg/g, respectively). According to the nutrient release and toxic metal content, FS_treated_ proved to be viable for agricultural applications; these findings illustrated that the “green” synthesis of nZVI-BC not only provides innovative and efficient Fenton catalysts, but also constitutes a novel approach for the utilization of sewage sludge, supporting overall process sustainability.

## 1. Introduction

Wastewaters are becoming an emerging threat because of the continuous release of various pollutants such as heavy metals, polycyclic aromatic hydrocarbons, polychlorinated biphenyls, and pharmaceuticals, etc. [1]. Water treatment resolves one issue, but it also produces sewage sludge (SS), which necessitates a distinct disposal and management strategy [2,3]. The annual production of sewage sludge in the 32 member countries of the European Environment Agency (EEA-32) is about 11.1 million in recent years, which equates to about 17 kg per person. Based on data reported in 2022, 34% of the sewage sludge was used in agriculture, 31% was incinerated, 12% was used in compost and other applications, 12% went to landfills, and 10% was used in other ways. Some countries predominantly send treated sewage sludge to land, while others incinerate it. The cost of the treatment plus disposal of sludge in European countries has been estimated to reach, on average, approximately EUR 200 per tonne of dry mass, according to the type of treatment and disposal [4]. Significant differences between the member states in sewage sludge management are reported; while the reuse of nutrients in agriculture (land spreading or composting) is the most common practice in Spain, Ireland, Finland, Hungary, and Cyprus, incineration is mainly applied in the Netherlands, Belgium, Germany, and Austria [5]. Around 35% of SS is used as a fertilizer in the United States. In Japan, 70% of SS is managed by incineration. More than half of the SS in South Korea is dumped into the sea [6]. Worldwide sludge production will continue to grow due to increasing urbanization and industrialization. Additionally, more strict environmental protection policies will influence upgrading old and building new wastewater treatment plants.

Until now, the focus of most research in the field of wastewater has been on technical aspects and improvements in water quality while minimizing the impact on the environment and human health. However, recent studies have shown that excluding broader social issues that affect the adoption of sustainable solutions prolongs the resolution of global environmental problems, as well as unjust health and social conditions in certain parts of the world. Therefore, increasing attention is paid to the social aspects of wastewater management strategies. This primarily refers to the wider public’s acceptance of the possibility of using purified wastewater and sludge generated during the application of the treatment process for various useful purposes.

Wastewater treatment plants (WWTPs) need to become one of the crucial elements of the regional bioeconomy, mainly through energy (waste to energy) and matter (nutrients-energy-water) recovery as an element of sustainable development. Today, in terms of circularity, both wastewater and the obtained wastewater sludge are concerned as resources. Currently, WWTPs should be an area where these assumptions can be successively implemented through closed water and wastewater cycles, the recovery of valuable substances (nitrogen, phosphorous, cellulose, humic acids, fatty acids, enzymes, and polysaccharides), the recovery of products generated in technological processes (fertilizers and biochar), and the recovery of energy and heat [7].

Sewage sludge consists of a heterogeneous mixture of useful and harmful substances such as heavy metals, persistent organic pollutants, pathogens, organic matter, and macro- and micro-nutrients [8,9]. Conventional methods used for SS management are no longer viable because of strict regulations, the lack of available space, and the increasing environmental and health problems caused by the presence of pathogenic agents, pharmaceutical substances, hormones, heavy metals, and persistent organic pollutants. Sewage sludge pyrolysis is considered an acceptable method from economic and ecological perspectives for the beneficial reuse of sewage sludge. This method has many advantages such as reducing sludge volume by 80%, the removal of pathogenic agents and hazardous compounds from SS, metals being immobilized in a solid residue, thus reducing their leaching, and organic and inorganic fractions being immobilized in a stabilized form of pyrolytic residues (biochar) [10]. There are many benefits of biochar (BC) production: energy production, sustainable waste recycling, carbon sequestration, improvement in soil quality, plant development, and mitigating greenhouse gas (GHG) emissions [6,11,12]. Many studies have proven that the pyrolysis of sewage sludge to biochar can be more effective than the application of raw sewage sludge in reducing the environmental risks of the potentially toxic metals involved in land application [13,14]. 

Besides the above-mentioned uses, considerable research attention is attributed to biochar as a carbon-based catalyst for different environmental applications. It can function as a catalyst in advanced oxidation processes to facilitate the generation of sulfate radicals, superoxide radicals, and hydroxyl radicals with low energy requirements and transition metals for degrading refractory pollutants. This may be attributed to the large amount of nutrients (N and P), metals (Ni, Zn, Mg, Ca, Ti, Cu, Fe and Al), O, S, Si, and C, which are always involved in the preparation of different heterogeneous carbonaceous catalysts. To date, the application of modified sewage sludge biochars as catalysts has been investigated in catalytic reactions (e.g., photocatalysis, H_2_O_2_/persulfate activation, and Fenton-like/ozonation reaction) for the removal of organic contaminants [12].

Owing to a large pore volume, specific surface area, and various active functional groups, minerals with aromatizing structures, and not being biodegradable by microbes, BC possesses high chemical and biological stability compared with other carbon sources. These adjectives indicate its application as a waste-derived low-cost BC catalyst for the abatement of recalcitrant organic pollutants from wastewater. However, in general, the removal ability of raw biochar to pollutants is still limited. 

Furthermore, biochar could be sustainably used to form carbon–iron composite materials, using external iron sources, for treating various recalcitrant organic pollutants in advanced oxidation processes.

Nanoscale zero-valent iron (nZVI), as a potential alternative source of Fe^2+^, has been successfully used to activate hydrogen peroxide for the degradation of various pollutants [15,16,17]. Green synthesis of nZVI, using plant extract, attributes high activity, energy saving, and lower cost for large-scale implementation [15]. However, nZVI tends to aggregate into forming microscale particles, resulting in diminished reactivity. Considering its large surface area, porous structure, and cost effectiveness, biochar can represent a great supporting material to stabilize nZVI and enhance its catalytic ability [18].

The objective of this study was to examine the degradation performance of a biochar–iron composite material synthesized using sewage sludge from wastewater treatment, which would otherwise be land filled. The work intended to assess an alternative approach that may utilize the large volumes of sewage sludge from wastewater treatment for beneficial use in environmental remediation, having significance in the reuse of resources. The circular economy concept is gaining traction due to its capacity for environmental sustainability, which includes resource utilization efficiency, nutrient recycling, and cascade use, in which all materials at each stage of the process are recognized as valuable resources instead of wastes. Unlike previous papers, where the water treatment was only assessed, the authors proposed a new approach to wastewater treatment, combining solutions for both water and sludge.

In this research, one possible sewage sludge valorization pathway was proposed, relying on circular economy principles.

The work included several consecutive steps towards exploring the possibility of closing-the-loop in wastewater treatment and sludge management (Figure 1): 

(1) Sewage sludge conversion to biochar (BC) through pyrolysis; 

(2) ‘Green’ synthesis of nZVI supported on BC; 

(3) Evaluating the nZVI-BC, as a catalyst in the Fenton process, for wastewater treatment containing Reactive Blue 4 (RB4) as a model substance due to its complex and hardly degradable anthraquinone structure. The use of nZVI-BC promotes environmental friendliness, wide applicability, and simple, rapid degradation of organic compounds; 

(4) Remained Fenton sludge (FS) was subjected to ignition, forming FS_treated_. The characteristics of this newly produced material were assessed to investigate the degree of retained catalytic properties (mainly iron form and content); 

(5) FS_treated_ was returned to the Fenton process to examine its efficiency as a recycled catalytic material. In this way, sustainability of the entire treatment could be achieved; 

(6) Assessing the potential of FS_exhausted_ as soil amendment/conditioner in comparison to BC, through elemental analysis and nutrient-release experiment, as a potential method to close the loop of the exhausted catalyst stream.

## 2. Results and Discussion

### 2.1. Characterization of Raw and Modified BC-Based Materials

The morphologies of BC, nZVI-BC, and FS_treated_ were analyzed by SEM and are presented in Figure 2. BC was characterized by a smooth surface and porous structure (Figure 2a). There were many tiny pores in the interior and on the surface of the original biochar, which provide sites for the adsorption of nZVI particles. The particle size of BC was differentiated ranging from 600 nm to 2.15 μm. In the absence of BC, the “green” synthesized nZVI particles were spherical and sized from 30 to 825 nm (published in [19]). When introduced to BC, the nZVI particles were randomly located and roughened the BC surface. nZVI-BC particles were twisted and serpentine and abundant pores could be observed (Figure 2b). The particle size ranged from 68 to 521 nm. The initial Fe^3+^ concentration, initial substrate concentration, temperature, pH, and stirring rate of the reaction medium were dependent on the factor for controlling the nanoparticle size and size distributions according to Turabik and Simsek, 2017 [20]. They established that Fe^3+^ concentrations lower than the stoichiometric requirement contribute to the synthesis of larger nanoparticles. The temperature increase showed similar effects, as well as the increase of the stirring rate over 400 rpm. The surface of FS_treated_ was coarser with rough aggregates on the surface, which could be related to the corrosion of nZVI particles and dye/Fe-oxides via redox reactions (Figure 2c).

The EDS results (Table 1) of the elemental qualitative chemical analysis confirmed the presence of carbon, oxygen, chlorine, calcium, and phosphorous atoms, in addition to iron, on the surface of the synthesized nanomaterial based on the obtained intense peaks of detected atoms. The C signals, as well as macronutrient content, originate from the biochar itself and from polyphenol groups and other C-containing molecules in the oak extract. Moreover, FeCl_3_ utilized in the synthesis of oak-nZVI must be accountable for the appearance of the Cl element. The Ca element is attributed to the oak extracts because Ca is indispensable for plant growth and exists in every living plant cell. The Fe peak in the EDS spectra (Figure not shown in the paper) and the uniform Fe distribution in the SEM element mapping (Figure 2b) further verified the nZVI synthesis on BC. 

Concerning the EDS analysis of FS_treated_, it can be concluded that combustion processes influenced the carbon, which decreased to 19.84%, in favor of other elements contributing to the formation of minerals. Regarding Fe and O content, their uniform compositions did not change significantly, which led us to the conclusion that FS_treated_ still can be used as a catalyst. The K and P contents in FS_treated_ were assessed to be 3.62% and 3.85%, respectively. These values are in accordance with other authors who dealt with sewage sludge biochar [21].

Dynamic light scattering (DLS) is one of the most popular methods for particle size determination, especially for nanoparticles. According to the particle size distribution curve, relative to the total number, most of the nZVI-BC particles were about 65 nm, and for FS_treated_ most particles were in the range between 100–200 in size (Figure 3). These changes in particles size could be the consequence of combustion processes under oxide conditions, where the formation of various Fe-oxides was observed. Newly formed particles were larger in diameter and shifted from the nanoparticle to microparticle region [22,23,24,25]. Peak intensity was higher for the nZVI-BC sample as these particles were more uniform in size and were in a narrower range upon controlled synthesis. After combustion (FS_treated_), more size-diverse particles were present with the main range between 125–250 nm (68%), but also with a significant portion of particles all through the range of 250–2500 nm (32%).

During pyrolysis, chemical conversion takes place. At low temperatures (up to 400 °C), cellulose, hemicellulose, and lignin decompose. At 300–400 °C, which was the preferred temperature in this experiment, volatile components in sludge were lost due to their conversion into tars or gaseous contents. Further carbon loss occurred due to the loss of carboxyl C and O-alkyl C groups. Other non-metals (O, N, S) had similar decreasing trends. The biochar produced in this temperature range showed more aliphatic functional groups on the surface and less aromatic structures because aromatization is expected at higher temperatures. The Fe species in the sludge biochar, at lower temperatures, remained almost unchanged with no crystallinity present. Fe reduction and crystals formation were favored at higher pyrolysis temperatures. Fe_3_O_4_ phase structure is expected to form in BC at moderate heating temperatures (≥400 °C), which was achieved in this research.

The outcome of XRD analysis is provided in Figure 4 to reveal the possible crystal structures of the pristine BC and nZVI-BC and FS_treated_. It is observed that α-Fe0 was confirmed based on a broad peak centered at 44–45° 2θ [26] and an XRD peak at 2θ = 20–25° was assigned to the amorphous structure of graphite of biochar [27]. Both characteristic diffraction peaks of Fe^0^ in the XRD spectrogram of nZVI-BC indicated successful synthesis of the composites. For both nZVI (previously published in [19]) and nZVI-BC, the appearance of a characteristic diffraction peak of Fe^0^ at 2θ of 44.8° illustrated the successful loading of Fe^0^ on BC [16]. However, this diffraction peak on nZVI-BC was much broader than that on nZVI, suggesting the decrease of crystal size and the intensified dispersion of nZVI after loading. Many researchers indicated [27,28] that nZVI can be transformed into Fe_2_O_3_, FeO, or Fe_3_O_4_ during combustion processes, which was confirmed by XRD analysis (Figure 4). The carbon sheets (2θ = 20–25°) could efficiently prevent the iron oxide nanoparticles from aggregating. After regeneration, the nZVI immobilized on BC was oxidized to Fe_3_O_4_ or γ-Fe_2_O_3_, evidenced by the disappearance of the diffraction peak for Fe0 and the emergence of diffraction peaks for magnetite at 2θ of 35.4° and 62.5° 

### 2.2. DSD Model Evaluation and Fenton Process Optimization

To characterize the system under the influence of different process conditions, initial dye concentrations, initial catalyst concentrations, and initial hydrogen peroxide concentrations on the decolorization efficiency of Reactive Blue 4 synthetic dye solution, DSD statistical analysis was applied. As explained in Section 3.6, the basic scheme of the DSD experiment with three numerical factors consists of 13 experiments, in duplicate and two central points. The total number of experiments was 28 (Table 2). The results of removal efficiencies of RB4 are shown in Table 2, where the range of decolorization efficiency of 5.03%–92.96% was established.

Achieving maximum and minimum decolorization efficiency in the Fenton process was observed in a distinct set of process conditions, confirming the assumption that the RB4 removal process itself depends on the applied experimental conditions. To select the regression model that best approximates the obtained results, JMP stepwise regression analysis was applied, which generates many regression models with different numbers of parameters, considering the main factors and their two-factor interactions. From a large set of regression models, a small set of candidate models with different numbers of members was selected, all of which provide good data approximation. The final selection of the final model was based on the standard selection criteria: BIC (Bayesian Information Criterion), AIC (Akaike Information Criterion), and RMSE (Root Mean Square Error) indicators (Table 3) [29]. AIC and BIC indicators are very similar in form, but derive from different assumptions. The AIC indicator is presented as an indicator of the relative quality of statistical models for a given data set and its role is to select the model that produces the probability distribution with the smallest deviation from the true distribution. In other words, AIC evaluates the quality of an individual model in relation to all observed models, relying on good approximation and the simplicity of the model. On the other hand, the role of the BIC parameter is to remove inadequate data fitting. Lower values of BIC and AIC indicators indicate a better ability to predict the regression model. RMSE represents the standard deviation of residuals (prediction errors) and provides insight into how much data are concentrated around the line of best fit [30].

In addition to the standard selection criteria for the appropriate model, an additional technological criterion is that the models must contain a dye concentration factor, but also all other tested input parameters, which is extremely important from an engineering point of view in solving the observed problem. Therefore, the final model selection was based on the lowest values of BIC, AIC, and RMSE indicators, respecting the simplicity of the model as an additional criterion. High values of the coefficient of determination (R^2^) and the adjusted coefficient of determination (R^2^_adj_) were observed in all applied Fenton processes (Table 3), which implies a good approximation of the experimental data with the selected model, i.e., indicates the absence of over-adaptation of the model to the data. The obtained values of descriptive factors R^2^ and R^2^_adj_ represent the percentage of data closest to the best-fit line, and indicate the fact that 98–99% of the variance for RB4 dye removal efficiency is explained by an independent variable, while the remaining 1–2% of the total variance is not covered by the model.

Figure 5 shows the 3D plots of the response surfaces of the examined two-factor interactions. In the first place, it is important to note that reaction medium was acidic (pH = 3.45), which positively affected the Fenton reaction, resulting in the formation of highly reactive hydroxyl radicals [17]. It should be noted that no pH correction was performed. Comparing the effect of variation in the initial concentration of hydrogen peroxide and catalyst (x and y axes), in terms of the response–removal efficiency (z axis; Figure 5a), an increase in decolorization efficiency was observed with an increase in both factors. 

It is considered that the highest concentration of nZVI-BC catalysts provides sufficient active sites for hydrogen peroxide decomposition and the production of the desired radicals [31], as indicated by the results of tests 7 and 20 (Table 2) with achieved efficiencies of 90.05% and 92.96%, respectively. When observing the influence of the initial concentration of hydrogen peroxide and pollutant (x and y axes; Figure 5b), a difference between the upper and lower tested dose of oxidant was noticed. Namely, at 10 mM H_2_O_2_, a significant decolorization of the aqueous solution was achieved with a decrease in the concentration of RB4 dye. This trend of declining efficiency of the Fenton process has been observed in other studies [32,33], and this phenomenon is due to the possible competitive adsorption of dye molecules on the surface of the solid catalyst, thus occupying available sites for reaction with hydrogen peroxide [34]. Figure 5c shows the influence of the initial concentration of the catalyst and RB4 dye, where a significant increase in treatment efficiency was observed, which is assisted by the development of reactions between RB4 dye molecules and nZVI-BC. However, as previous tests on the efficiency of RB4 dye removal by sorption (35% and 70% for 20 mg/L and 100 mg/L, respectively) have shown, it can be concluded that both processes (oxidation and sorption) can take place simultaneously, which, in optimal conditions, leads to high treatment efficiency. The statistical model within the Fenton process proposes a high dye-removal efficiency of 95.02% under the following optimal conditions: dye concentration of 50 mg/L, catalyst concentration of 200 mg/L, and hydrogen peroxide concentration of 10 mM. The mean value of the efficiencies obtained was 98.48%, with a standard deviation of 0.24. The obtained results are promising from the aspect of overcoming the limiting factors of the homogeneous Fenton process, such as the need to maintain the operating pH value within the narrow range of 2.8–3.5 to achieve the maximum catalytic activity of iron as a catalyst and prevent its deposition as hydroxide.

Fenton sludge can be further utilized to synthesize heterogeneous catalysts/biochar via pyrolysis and hydrothermal carbonization [35,36,37,38].

To examine the performance of the new catalyst, decolorization of RB4 dye at optimized conditions was performed once more. The five replicates’ efficiencies were 99.56% ± 0.15. This small set of experiments represents the proof of successful utilization of Fenton sludge as a new iron-based heterogeneous catalyst. It is one of the potential options to address sustainability issues. In this way, the reuse routines have a direct influence on minimizing the yield of Fenton sludge. The contribution to circular economy aims was achieved by the reuse of recycled waste material as a new product, as opposed to discarding it.

### 2.3. Treatment and Characterization of Real Effluent of RB4 Dye under Optimal Conditions of Fenton Process Catalyzed with nZVI-BC

After the treatments, the characterization of the treated effluent was performed. The results of the physico-chemical characterization of the effluent before and after treatment are shown in Table 4.

Increased conductivity after the treatment indicates the formation of degradation products and the release of certain inorganic ions, which can originate from the dye molecule itself. These inorganic ions compete with Fenton catalysts, since they can act as traps for hydroxyl radicals and thus contribute to the reduced efficiency of the applied Fenton treatment.

The BOD value of real effluent was below the detection limit, which confirmed the fact that real effluent is non-biodegradable, and it is not possible to apply biological treatment to achieve high efficiency of dye degradation. The increased value of BOD after treatment in the treated samples indicated the fact that several degradation products were formed, which confirms the assumption that removing dye from the solution and decomposition of the chromophore group do not necessarily mean its complete oxidation to CO_2_ and H_2_O. The derived conclusion is in accordance with the results of determining the TOC and COD values. Namely, the degree of mineralization of RB4 after the homogeneous Fenton process was determined by measuring the degree of removal of the TOC value (93.7%) and the degree of reduction of the COD value (62.5%) under optimal process conditions of the decolorization reaction. For a set of five samples under optimal conditions, the spectra of the final effluents after decolorization were recorded (Figure 6). Spectrum analysis concludes that converging the conditions to the optimal ones reduces the absorbance not only in the selected visible region, but also in the UV region. This indicates that not only the chromophore group is destroyed, but also the degradation of aromatic structures, which are dye molecules.

There was a noticeable decrease in the absorption at wavelengths in the range of 250–300 nm, which corresponds to the absorption maxima of the components containing the benzene ring, aromatic and dichlorotriazine groups, and naphthalene compounds [26]. However, below 250 nm, the appearance of new degradation products was observed, which was indicated by the increase in conductivity after paint treatment in optimal conditions.

### 2.4. FTIR Analysis of Effluents

The results of the FTIR analysis are shown in Figure 7. This analysis was performed for the synthetic dye solution before treatment as well as for the effluent after treatment.

For the dye RB4, the initial effluent indicated the presence of the expected groups, such as groups with double bonds, primarily carbonyl groups with oscillation at 1720 cm^−1^ with oscillation type ν (C=O), and different aromatic structures with halogen atoms (in this case it is Cl) with the frequency of the absorption band at 680–725 and 750–810 cm^−1^ δ (CH) out of the plane, such as the aromatic ring frequency at 1450 ± 10 cm^−1^ with the ν (C=C) and ν (C-C) skeletal types of oscillations. Amino groups with aryl residues in the frequency range 1280–1350 cm^−1^, ν (C-N), as well as aromatics containing nitrogen in the structure (pyridine-type structure) with a frequency of 1200–1300 cm^−1^ ν (N-O) were also identified [39]. After the treatment, the recorded spectra showed significantly lower frequency maxima, which indicated the mineralization of the solution as well as the degradation that took place in the direction of creating smaller molecules and reducing aromatic structures. Peaks in the range of 1000–3300 cm^−1^ are attributed to vibrations of O-H bond stretching in carboxylic acid molecules. The second maximum at 1617 cm^−1^ corresponds to the tensile vibrations of the C=C bond of the polyphenolic components. The maximum at 1031 cm^−1^ indicated the presence of C-C tensile vibrations, as well as C=C, C-O-C, and O-H vibrations. The maximum at 441 cm^−1^ is attributed to -S-S- tensile vibrations. The identified peaks of characteristic functional groups were been confirmed by the author in [40].

### 2.5. Mechanism of RB4 Removal by nZVI-BC

Qualitative gas chromatographic/mass spectrometric analysis was used to identify the nature of the degradation products of the treated effluents. The identified compounds, which showed the strongest intensity on the chromatogram, are shown in Table 5. After treatment, a small number of compounds were identified, indicating an almost complete mineralization by this enhanced oxidation process. The results confirmed that at the very beginning of the degradation process, the chromophoric groups of the corresponding dye were broken. On the chromatogram of the effluent after the treatment of the RB4 dye solution, only one peak of significant intensity was obtained, for which three degradation compounds were proposed based on the analysis of the results. The proposed compounds unequivocally indicated that cleavage of the anthraquinone part of the molecule certainly occurred in the later stage of the reaction. Namely, it is assumed that due to its complexity and the difficulty of finding the place of “attack” by hydroxyl radicals, this structure is the most difficult to cleave, and its residues were the only ones identified after 60 min of treatment under optimal conditions. Since nZVI particles are good electron donors and dye molecules are excellent electron acceptors, Fe^0^ nanoparticles were reduced to Fe^2+^ and Fe^3+^ ions in the aqueous medium and the hydroxyl and/or hydrogen ions generated during the reduction process reacted with dye molecules to induce the breaking of the chromophore. The initial oxidation product of the RB4 dye (1-amino-9,10-anthracenedione) as well as triazine compounds were not identified by mass spectrogram, indicating the dye’s degradation to phthalates, phenols, and organic acids [41]. Further decomposition of the dye molecules proceeded towards the creation of simpler aromatic and aliphatic intermediates, as well as products created by the “opening” of aromatic rings, leading to oxidation of the final products CO_2_, H_2_O, and inorganic salts. As analysis was performed for the sample under optimal conditions and after one hour of reaction, the obtained results clearly indicated that advanced degradation phase did actually occur, and that most of the dye molecules were transferred to end products, with some remains of the organic structures.

### 2.6. Stability of nZVI-BC

To confirm the stability of nZVI-BC, iron leaching as well as the aging effect were investigated. After the application of optimal conditions ([H_2_O_2_] = 10 mM, [RB4] = 50 mg/L and [nZVI-BC] = 200 mg/L, pH = 3.2.) on RB4 decolorization, the concentration of leached Fe was measured within 60 min. The obtained leached concentrations of iron are presented in Figure 8. All the leached iron concentration were ≤5.5 mg/L. The exhibited low iron leaching, no more than 2.5%, indicated the good structural stability of the catalyst. The leached iron concentration of the ZVI-based catalysts was reported to vary from 0.01 mg/L to 240 mg/L, depending on catalyst properties, pH, H_2_O_2_ concentration, reaction time, and catalyst dosage [42].

To investigate the aging effect of nZVI-BC, we let the prepared material age one month in a closed vial at 4 °C. Then, it was used again for the decolorization experiment under optimal conditions (Figure 9). The efficiency of freshly prepared nZVI-BC on dye degradation during optimized conditions was 98.48 ± 0.24. The oxygen could convert Fe^0^ to ferrous or ferric oxide, leading to a passivation layer forming on the nZVI surface. To study the mentioned effect, the same experiment was performed after a month with the same synthesized material. The removal efficiency of the so-called aged nZVI-BC decreased insignificantly by 4.8%, after one month, in comparison with the freshly prepared catalyst. It could be concluded that the aging of nZVI-supported on biochar insignificantly contributed to the delay of the formation of the passivity layer. The phenomenon could be explained by the fact that nZVI particles were synthesized in the pores of BC, or in the inner surface, thus preventing iron nanoparticles from interacting with air and delaying the formation of iron oxides. Comparable results were confirmed by other researchers using biochar as supports [43,44].

### 2.7. Fenton Sludge as a Nutrient Resource

The results of the chemical analysis of FS_treated_ determined by ICP-MS, as well as the comparison with emission limit values for biosolids from municipal wastewater treatment according to the regulation of the Republic of Serbia (“Official Gazette of the RS” No. 67/11, 48/12 and 01/16) and EU Directive 86/278/EEC) [45,46] on sewage sludge application in agriculture, are shown in Table 6. It can be observed that the FS_treated_ is rich in different minerals. Mineral content is an important parameter for biochar that is produced for land applications. This implies the potential for the recovery of plant minerals. 

According to the results of chemical analysis, FS_treated_ contained various amounts of different kinds of elements. Even though some of the concentrations had excessive concentrations, if the biomass was mixed with something else (for instance carbon reach biomass), it was possible to reach permissible element concentrations. 

One can conclude that biochar originated from sewage sludge generally contains high concentrations of metals, which remain in the final biochar product following pyrolysis. Many of the identified elements are not regulated if the biochar is applied on land but increasing concerns might lead to the formation of new regulations. The currently regulated heavy metals only include Cd, Cr, Cu, Hg, Ni, Pb, Zn, and As. For the comparison of results with permissible concentrations of regulated elements in Serbia, 86/278/EEC Directive [46] values were chosen as well. The heavy metal content of the FS_treated_ met the requirements of emission limit values for biosolids from municipal wastewater treatment according to regulation of the Republic of Serbia [45] for use in agriculture and other uses such as landfill covering, greening of parks, as soil conditioner/amendments on which agricultural crops and livestock will not be grazed for at least one year, and for filling depressions (landscape improvement) etc. Additionally, the biochar’s heavy metal content fulfills the conditions for the maximum permissible concentrations of Directive 86/278/EEC for land applications.

Since Directive 89/278/EEC is outdated, each country develops and implements its own regulations. However, those regulations vary to a great extent. This makes it difficult for the international biochar market (to sell or buy biochar abroad). Therefore, the development of international regulations would benefit all countries. There are multiple initiatives for regulation development including European Biochar Certificate (EBC) developed by Ithaka Institute in Switzerland, International Biochar Initiative (IBI) standards, Australia and New Zealand Biochar Standard, and Korea Biochar [47].

However, most recently the European Biochar Certificate (EBC, 2021) [48] defined four application classes as EBC-Feed (Class I), EBC-AgroBio (Class II for organic products, EU regulations for organic compost), EBC-Agro (Class III, based on the German Federal Soil Protection Ordinance (BBodSchV)), and EBC-Material (Class IV, based on the Swiss Ordinance on Waste for the Production of Cement and Concrete) that meet the requirements for biochar’s application as a feed and feed additive in animal husbandry, for agricultural use as a fertilizer, and in industrial applications. Based on these values, one can conclude that FS_treated_’s heavy metal content does not meet the requirements for Classes I–III. An exception exists in the case of EBC Class IV, where the biochar’s heavy metal content overcomes 1.4 times the permissible concentration only for Zn, indicating its possible usage for industrial application. At the same time, if compared with requirements of IBI standards, characterized waste sludge biochar does not meet the requirements in terms of Cr, Ni, Cu, Zn, and Cd.

**Table 6 molecules-28-01425-t006:** Chemical analysis of the FS_treated_ material.

Element	Concentration, mg/kg	Directive 86/278/EEC [46]	Regulation on Emission Limit Values for Pollutants in Waters and the Deadlines for Their Reaching (“Official Gazette of the RS” No67/11, 48/12 and 01/16) [45]	EBC [48].				IBI
				EBC-Feed Clas I	EBC-AgroBio, Class II	EBC-Agro, Class III	EBC-Material, Class IV	
Cr	95	-	100–1000	70	70	90	250	93
Ni	55.01	300–400	60–400	25	25	50	250	47
Cu	197.49	1000–1750	700–1750	70	70	100	250	143
Zn	1045	2500–4000	1500–4000	200	200	400	750	416
As	4.71	-	15–75	2	13	13	15	13
Cd	1.89	20–40	2.5–40	0.8	0.7	1.5	5	1.4
Pb	16.90	750–1200	120–1200	10	45	150	250	121

For healthy plant growth, 16 essential elements are necessary and include primary (needed in a large amount), secondary, and micronutrients [49]. All the elements, however, should correspond to the plant need as both deficiency and excessive amounts can cause serious damage to plant development.

When it comes to nutrient-release assays, the leaching experiments of K, Ca, Mg, Na, and P from regenerated biochar were carried out with the same biochar for five cycles with a total duration of 10 days under the experimental conditions given in Section 3.7.2. The cumulated amounts and release kinetics of K, Ca, Mg, Na, and P from the regenerated biochar are illustrated in Figure 10. It can be clearly noticed that the cumulative amount of released nutrients significantly increased with the number of leaching experiments. The highest cumulative amounts of released K, Ca, Mg, Na, and P were therefore observed at the fifth leaching cycle (6.40, 1.66, 1.12, 0.62, 0.48, and 58.2 mg/g, respectively) as shown in Table 7. These contents were, respectively, 58%, 90%, 93%, 63%, 129%, and 271% higher than the ones determined at the first cycle.

However, the release kinetics of nutrients from the biochar decreased rapidly over time.

For example, the calculated rates for Na, Mg, P, K, and Ca decreased by 94.5%, 89.7%, 64.8%, 96%, and 90%, respectively, between the first and the third cycle (Figure 10). The released rates of nutrients remained important even after five leaching cycles with values of 0.72, 0.03, and 0.03 mg/g/day for K, Ca, P, and Mg, respectively (Figure 10). Similar trends were reported by Ferjani et al., 2020 [50] when studying K, Ca, P, and Mg release from exhausted grape marc (EGM) biochar; Mukherjee and Zimmerman (2013) [51] when studying dissolved organic carbon and N and P release from five successive leaching assays of various biochars generated from lignocellulosic biomasses, and by Hadroug et al. (2019) [52] during their investigation related to P and K release from raw poultry manure (RPM) and RPM-derived biochars at temperatures of 400 and 600 °C. However, the nutrient-release kinetics of regenerated biochar from this study were much lower than those observed for EGM and RPM biochars. These inquiries display a beneficial strong point in biochar’s agricultural applicability, enabling slow nutrient release to crops, and at the same time disallowing groundwater pollution. The explanation for the higher kinetic rates of K and P compared with other macronutrients could be attributed to their higher concentrations, as well as the transformation of those elements into a more water-soluble form through dissociation mechanisms. For this reason, the biochar-borne K was more easily leached by distilled water. The biochar in this work was prepared at 400 °C and Li et al., 2018 [53] pointed out that pyrolysis temperatures above 450 °C form less-available P-forms such as hydroxyapatite and oxyapatite. 

Accordingly, the leached K percentage (calculated by Equation (3)) increased significantly with the increase in the number of leaching assays. Indeed, it increased from about 77% at the first assay to more than 84% at the third cycle and reached 89% at the fifth one. However, the maximal leached percentage of P (see Equation (3)) observed at the fifth cycle was only 11% of its total content in the biochar. Comparable results were found in the paper of Hadroug et al. (2019) [52] regarding P and K release from raw poultry manure-based biochar. This result implies that P is found in the form of less-available phosphorous such as apatite, hydroxyapatite, tricalcium phosphate, and calcium–iron phosphate precipitates such as whitlockite and phosphate adsorbed onto the calcium carbonate surface [54,55]. Concerning Ca, which showed a significant cumulative leached percentage (from 53% to 100%, for first and fifth cycle, respectively), this finding indicated the transformation of the possible existing CaCO_3_ into more water-soluble Ca(HCO_3_)_2_ [56], which can be quickly absorbed by the crop. It prevents the overgrowth of crops, hardens the fruit, prolongs the storage period, promotes the absorption of phosphoric acid, and helps the crops to accumulate nutrients. Calcium in its water-soluble form is efficient and effective in natural farming. The Mg’s cumulative leached percentage from 11% to 22% demonstrated the existence of slightly water-soluble MgO, whose attraction forces are strong enough to be disassembled when reacted with water. The same could be concluded for Na. Based on the discussions presented in this section, it can be concluded that the use of exhausted biosorbents loaded with metal ions as soil fertilizers has several advantages, such as (i) simultaneously solving two environmental problems: the removal of recalcitrant organic compounds from industrial effluents and the enrichment of soil with essential microelements, (ii) leading to an added-value product from a waste resulting from wastewater treatment, in agreement with the principles of circular economy, and can therefore be considered as a low-cost process. In support of that, Khan et al., 2020 [57] pointed out that biochar can be used as an amendment for compost stabilization of divergent biowastes, improving the quality of the final compost by increasing the concentration of plant-available nutrients, enhancing maturity, decreasing composting duration, and reducing the toxicity of the compost.

## 3. Materials and Methods

### 3.1. Materials and Chemicals

All sample analyses were carried out directly, without pretreatment, and the chemicals used during the laboratory tests were of analytical grade. Hydrogen peroxide (30%), NaOH (>98.8%), FeCl_3_ was obtained from Sigma Aldrich, while cc H_2_SO_4_ (>96%) was produced by J.T. Baker. Deionized water was used for the preparation of all working solutions within the desired concentrations. Commercial anthraquinone dye reactive Blue 4 (RB4) (CAS no. 13324-20-4, molecular weight, MW = 697.43 g/mol), was obtained from Sigma-Aldrich. Figure 11 presents the structure of the investigated dye.

### 3.2. Sludge Sampling and BC Synthesis

The sewage sludge originated from the wastewater treatment plant in a small settlement Kovilj near city of Novi Sad, Autonomous Province of Vojvodina, Serbia, (3.500 EC-equivalent per capita). The GPS coordinates are N 45.24355, E 20.01587. The procedure of biochar preparation was performed according to the work of Chen et al., 2020. [58]. Sludge was delivered to the laboratory and dried in an oven at 105 °C and then combusted in a furnace under an inert atmosphere (Nabertherm, Germany) at 400 °C. Biochar yield after combustion the was 57.53%. 

### 3.3. Synthesis of nZVI-BC Catalyst

The ’’Green’’ synthesis method of nZVI particles was conducted using fallen leaves from oak trees (Quercus Peatrea) growing in the National Park of Fruska Gora in Vojvodina, Serbia. The leaves were milled by using a kitchen chopper, then sieved by using a 2 mm sieve and pre-dried at 50 °C in an oven for 48 h. The amount of 3.7 g of the oak leaves was measured and carried to a 300 mL Erlenmeyer flask, to which 100 mL of water was added. Then the flask was put in a shaker bath at 80 °C for 20 min. The extraction procedure was performed according to Machado et al., 2013 [59]. Further preparation of nZVI-BC was carried out according to Mortazavian et al., 2019 [60]. The oak leaf extract was mixed with 0.1 M Fe (III) and biochar in a ratio of 3:1:1. The material thus prepared was stirred for 60 min at ambient temperature in an ultrasonic bath. The material was stored in the refrigerator until further use.

### 3.4. Characterization of BC, nZVI-BC, and FS_treated_

The morphology of the synthesized materials was examined by using scanning electron microscopy (SEM) (TM3030 (Hitachi High-Technologies, S-4700 Type II, Tokyo, Japan)) followed with energy-dispersive X-ray spectroscopy (EDS). EDS mapped the present elements on the nanomaterial surface at 15 kV acceleration (Brucker Quantax 70 X-ray detector system, Brucker Nano, Berlin, Germany, GmbH Germany, Berlin, Germany). XRD analysis was performed with Rigaku MiniFlex II desktop X-ray diffractometer with Cu Ka radiation, in the 2θ° angle range 0–90°. Particle size was obtained by dynamic light scattering (DLS) on Dynamic Anton Paar Litesizer™500. Parameters that were followed to analyze the particle size of the nZVI-BC and FS_treated_ were particle size distribution by intensity, volume, and number.

### 3.5. Physico-Chemical Analysis of RB 4 before and after Treatment

pH value, conductivity, and temperature were measured using SenTix^®^21 electrode. Chemical Oxygen Demand (COD) was determined from the potassium dichromate volumetric method—SRPS ISO 6060: 1994 [61]. The determination of biological oxygen demand (BOD) after 5 days at 20 °C was performed with the manometric method—H1.002, by using the instrument Velp Scientifica Italia, Lowibond and WTW. The determination of total organic carbon (TOC) in water was performed by LiquiTOC II (Elementar, Langenselbold, Germany), with the method SRPS ISO 8245:2007 [62]. The FTIR spectra of RB4 were obtained with a Fourier transform infrared spectra Thermo-Nicolet Nexus 670. The spectra were recorded in the range of 4000–6000 cm^−1^ in diffuse reflection mode, at a resolution of 4 cm^−1^. Gas chromatography–mass spectrometry (GC-MS, Agilent 7890A/5975C, Tokyo, Japan) qualitative analysis was applied in terms of intermediate identification according to the EPA 3510C method.

### 3.6. Definitive Screening Design

The efficiency of the applied treatment depends on several process parameters, which require the optimization of the entire process. To overcome the problem with a limited number of operational variables (caused by the fact that the number of experiments increases sharply when more variables are included in the experimental design), it is necessary to use statistical screening methods that will identify significant variables and eliminate irrelevant ones. For this purpose, it is possible to use a set of empirical statistical methods based on the application of quantitative data of appropriately designed experiments in order to determine the optimal conditions. Accordingly, a new generation of experimental design, Definitive Screening Design (DSD) was introduced in 2011 by Jones and Nachtsheim [63]. The principle of the DSD statistical method is based on the application of a numerical algorithm that maximizes the matrix determinant of the main effect model. The analysis conceived in this way is used to determine significant factors and to predict their two-factor interactions, but also to estimate the model equation under which will be determined predicted response values. This statistical method enables the application of a significantly reduced number of performed experiments with maximum precision [64]. The basic scheme of the DSD experiment with three numerical factors consists of 13 experiments, which with replication and two additional central points makes a total of 28 experiments. Investigated parameters and their experimental levels (lower, central, and upper) are presented in Table 8. JMP software (15.2.1. SAS Institute, Cary, NA, USA) was applied to generate the experimental matrix and to statistically analyze the obtained data. 

The tests were conducted by a series of experiments on a JAR test apparatus (FC6S Velp scientific, Italy). Dye solutions with a volume of 0.25 L were mixed in laboratory beakers at intervals of one hour, at 150 rpm. After mixing, the solutions were filtered through a membrane filter. After filtration, the absorbance (A) was measured at the wavelength for RB4 λ_max_ = 595 nm. Determination of absorption maxima (λmax) by recording the spectrum of the dye solution, as well as monitoring the change in absorbance during the experiments was performed using a UV-VIS spectrophotometer. The series of experiments included the examination of the following operational conditions on the efficiency of dye degradation: nZVI concentrations, hydrogen peroxide concentrations, pH, as well as the influence of the initial dye concentration in solutions.

Decolorization efficiency of the aqueous solution was obtained based on the following Equation (1):Removal efficiency [%] = ((A_0_ − A_t_)/A_0_) × 100%,(1)
where A_0_ is the initial absorbance of the aqueous solution before the Fenton treatment, whereas A represents the absorbance of the aqueous solution after treatment.

### 3.7. Fenton Sludge (FS) Reuse

After the applied homogenous Fenton treatment on dye decolorization, the obtained Fenton sludge was exposed to a further combustion process in order to reuse as a potential resource from two aspects. The reacted and used nZVI-BC was then collected, washed with 1l od deionized water, dried at 105 °C, and combusted at 550 °C for 30 min. After preparation, the obtained material was named Fenton-sludge treated (FS_treated_). The same characterization was performed as in the case of nZVI-BC. The first aspect of its reuse was the examination of its catalytic activity during heterogeneous Fenton. The second aspect was to use as a resource in terms of macronutrient composition, towards nutrient source usage as a soil amendment/conditioner. 

#### 3.7.1. Application of FS_treated_ in RB4 Decolorization

The same dye decolorization experiments were performed, but this time using heterogeneous Fenton catalyst made from Fenton sludge (from the previous experiment, Section 3.6) as the iron source, and at the optimized conditions [H_2_O_2_] = 10 mM, [FS_treated_] = 200 mg/L and [RB4] = 50 mg/L. Importantly, the pH correction did not have to be performed, because the experiments were carried out at pH = 3.75.

#### 3.7.2. Nutrient-Release Experiments

The examination of the release kinetics of macronutrients (P, K, Ca, Mg, Na) from the treated Fenton sludge, FS_treated_ was performed through successive batch experiments for an overall duration of 10 days. The procedure was followed by shaking (at 400 rpm) FS_treated_ at concentration of 10 g/L, in distilled water for two days. At the end of each two-day assay, the biochar was recovered using vacuum filtration with 0.45 µm filter papers, followed with drying at 40 °C for 16 h. The dried sample was reused again for the subsequent leaching procedure. The concentrations of macronutrients in the liquid leaching samples were analyzed by an inductively coupled plasma spectrometry with mass spectrometry (Agilent ICP MS).

For each leaching experiment “*n*”, the released amounts of P, K, Ca, Mg, and Na per gram of biochar “*q_n_* (mg/g)” were determined as follows:(2)$$q_{n}\;=\frac{C_{n}}{D}$$
where the *C_n_* is concentration of investigated nutrients (mg/L) at each successive assay, and *D* is used biochar dose (g/L).

Kinetic release rate of investigated nutrients *S_n_* (mg/g/day) for a given leaching experiments “*n*”, was determined as follows:(3)$$S_{n}\;=\frac{q_{n}}{\begin{array}{l} t \end{array}}$$
where *t* is the contact time or the leaching assay duration.

The cumulative leached percentages of investigated nutrients *γ_n_* after leaching experiments were calculated as follows:(4)$$\gamma_{n}\;=\frac{\sum q_{n}}{\begin{array}{l} M_{BC} \end{array}}$$
where, *M_BC_* (mg/g) are nutrients’ contents in the used *BC*.

## 4. Conclusions and Outlook for Long-Term Sustainability

Conclusions retrieved from the experiments of this work are the following:The applied DSD methodology indicated very low concentrations of the obtained nZVI and hydrogen peroxide were required for almost maximum decolorization efficiency. The results are promising from the aspect of overcoming low operating pH value in terms of its adjustment.The valorization of the obtained Fenton sludge was performed by reusing it as a new heterogeneous Fenton catalyst (FS_treated_) at the same optimized conditions as for homogenous nZVI-BC catalyst. The almost 100% degradation efficiency of RB4 was obtained, with no pH correction, because the treatment took place at pH = 3.75.Nutrient-release experiments demonstrated the nanomaterial’s high capability to supplement agricultural fields with macronutrients (K, Mg, Ca, P, Na) through their release over excessive periods with slow rates. This is the principal key for optimal agricultural plant growth and soil maturing as well as for reduced environmental pollution risks by these elements.

The novelty of this paper is reflected by attempt to close the loop in wastewater treatment, by recycling its own sludge (trough generated catalyst) in the wastewater line, polluted with recalcitrant organic compounds. Additionally, wastewater treatment process residues were further characterized in terms of elucidating nutrient-release content and the potential to act as soil amendments enforcing a zero-waste principle.

Future research must concentrate on the implementation of biochar-based catalysts in both simulated and real effluents in order to extend the life span of the catalysts and broaden their applications; establishing the nexus between soil properties and produced organic soil amendments; and enforcing science for a policy approach to develop the market potential of generated soil amendments. 

## Figures and Tables

**Figure 1 molecules-28-01425-f001:**
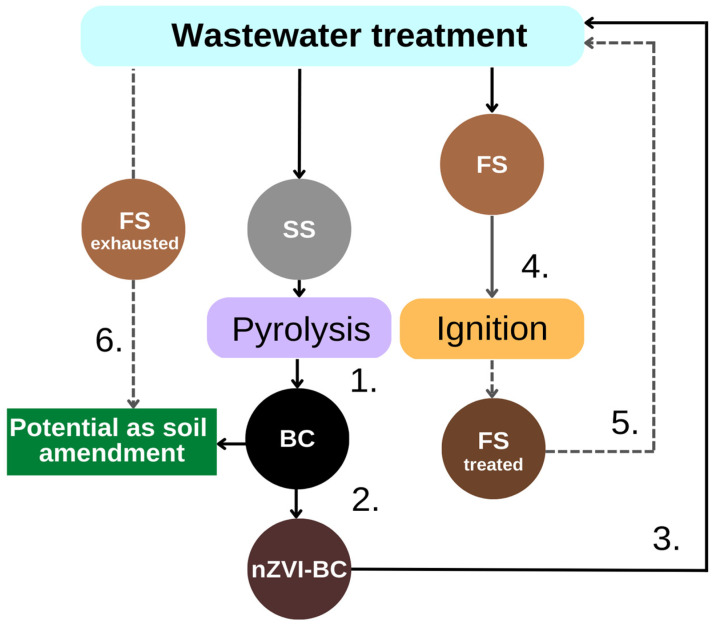
Schematic presentation of the phases of the conducted experiments (1. Pyrolysis, 2. Synthesis of nZVI-BC; 3. nZVI-BC application as Fenton catalyst; 4. ignition of Fenton sludge (FS) to form FS_treated_; 5. FS_treated_ application as Fenton catalyst; 6. FS application in nutrient release experiments after ignition).

**Figure 2 molecules-28-01425-f002:**
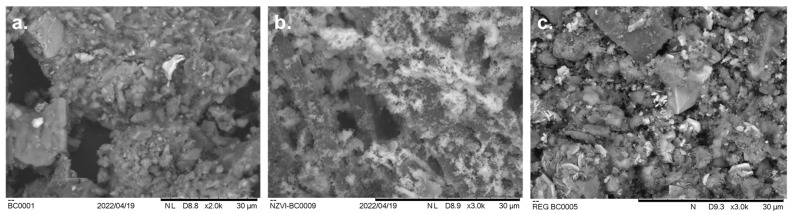
SEM images of bare (**a**) BC, (**b**) nZVI-BC, and (**c**) FS_treated_.

**Figure 3 molecules-28-01425-f003:**
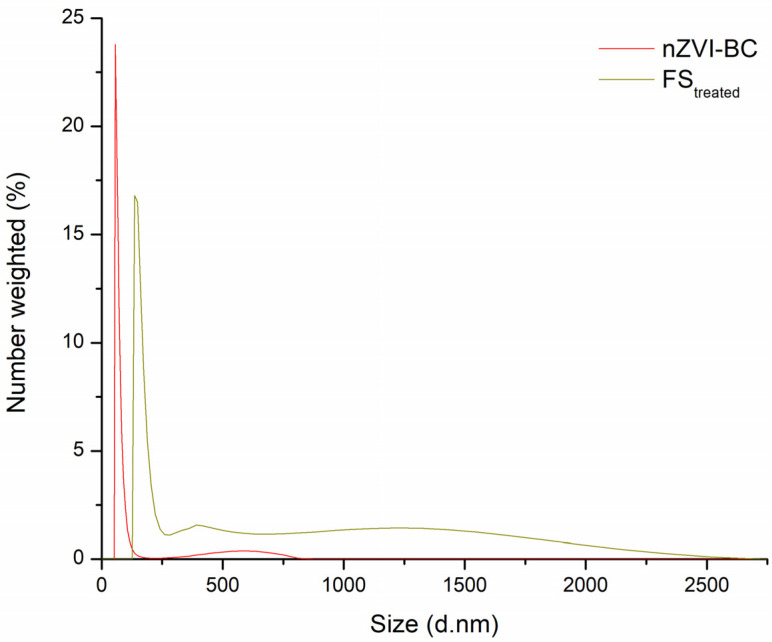
Particle size distribution curve for nZVI-BC and FS_treated_.

**Figure 4 molecules-28-01425-f004:**
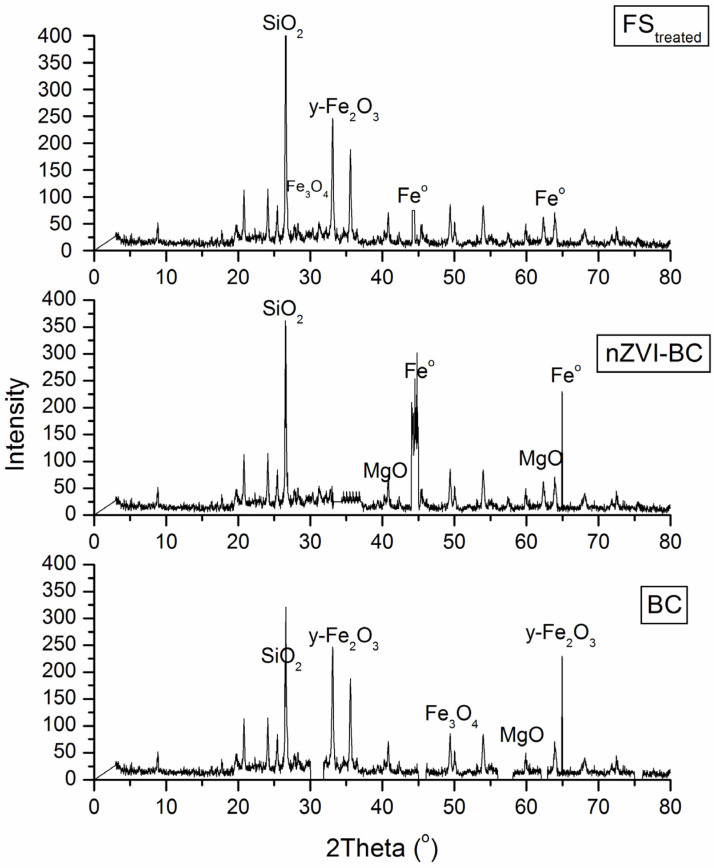
Difractograms of BC, nZVI-BC, and FS_treated_.

**Figure 5 molecules-28-01425-f005:**
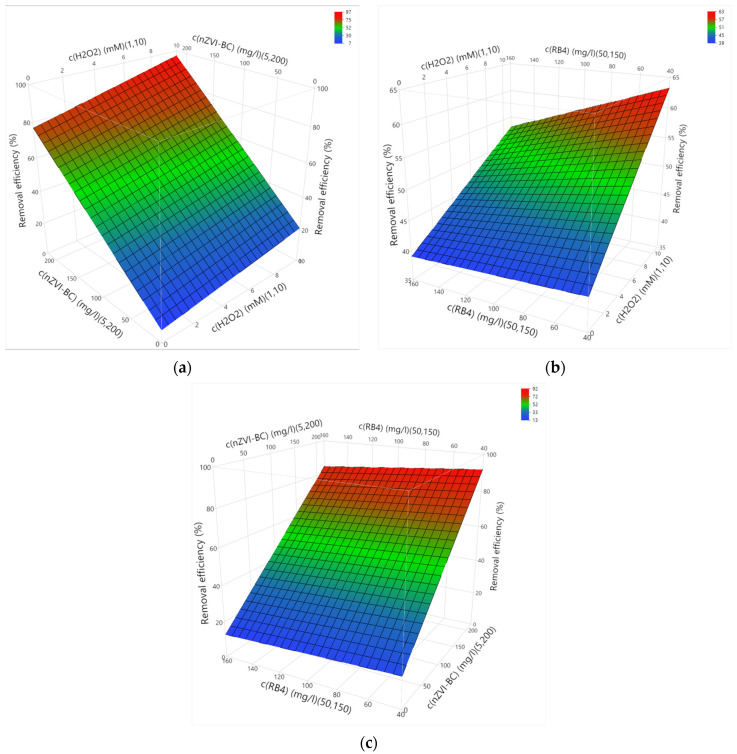
3D graphics of the response surface of statistically significant interactions: (**a**) H_2_O_2_ concentration and catalyst concentration; (**b**) H_2_O_2_ concentration and dye concentration; (**c**) dye concentration and catalyst concentration.

**Figure 6 molecules-28-01425-f006:**
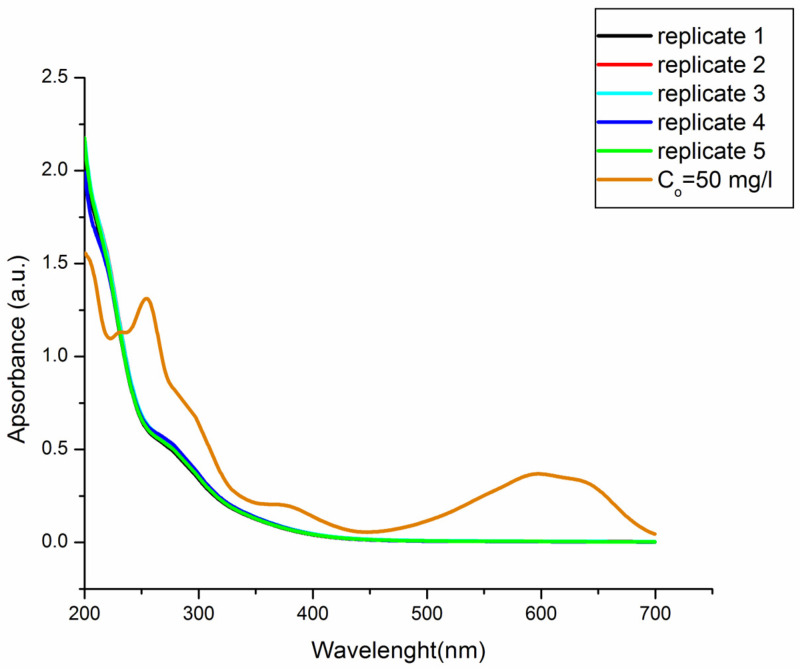
Spectra of the final effluent after decolorization under optimal conditions (performed in five replicates).

**Figure 7 molecules-28-01425-f007:**
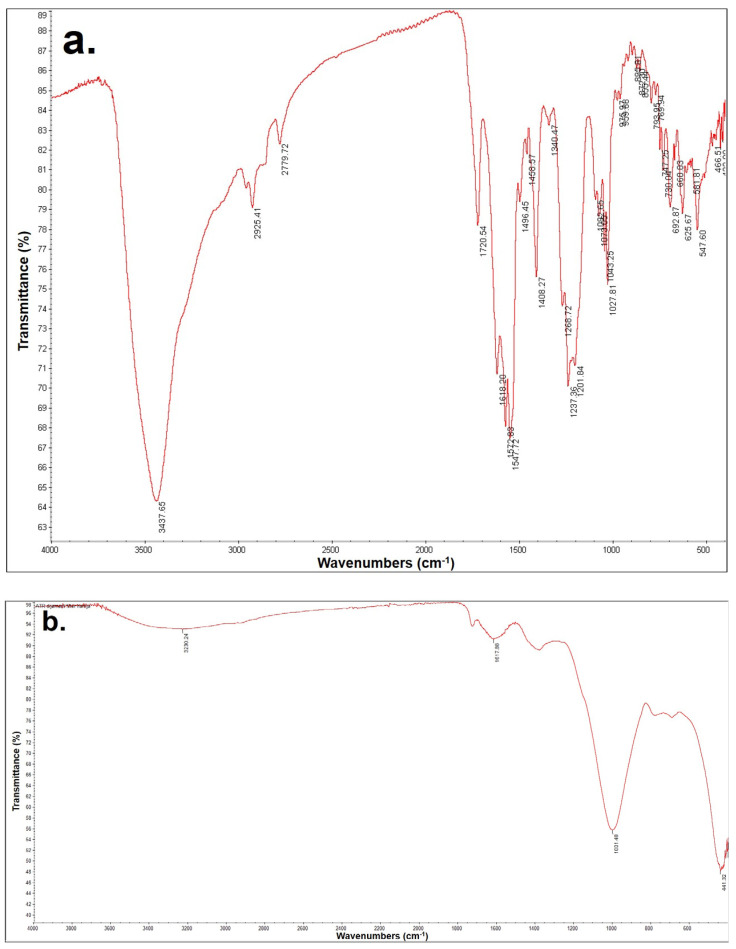
FTIR spectra of the (**a**) RB4 solution before treatment, (**b**) RB4 effluent after treatment.

**Figure 8 molecules-28-01425-f008:**
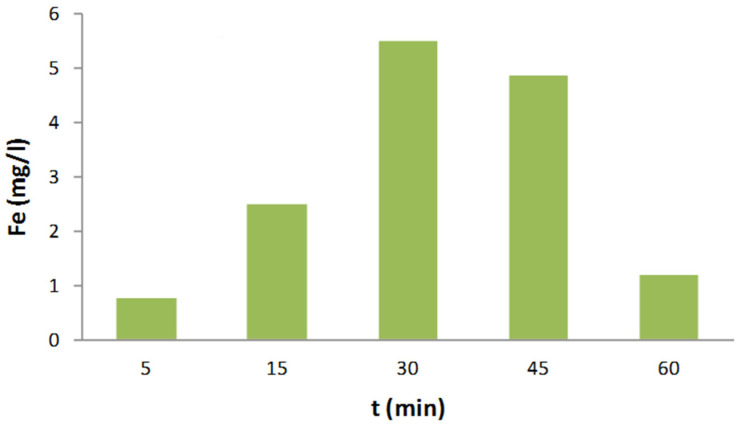
Leaching iron concentration after the applied optimized Fenton conditions within 60 min.

**Figure 9 molecules-28-01425-f009:**
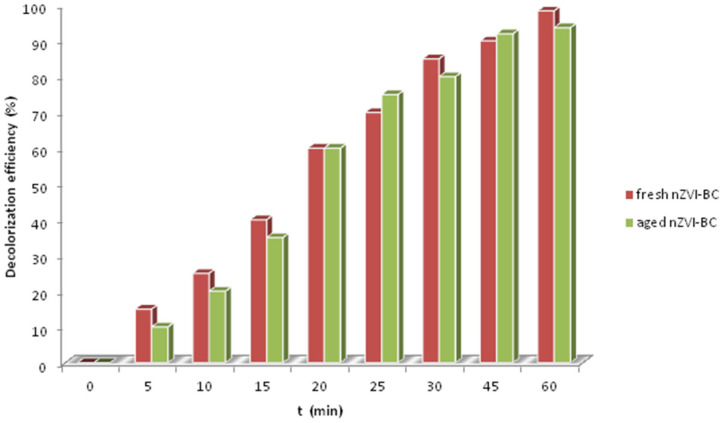
Effects of aging nZVI-BC on RB4 decolorization.

**Figure 10 molecules-28-01425-f010:**
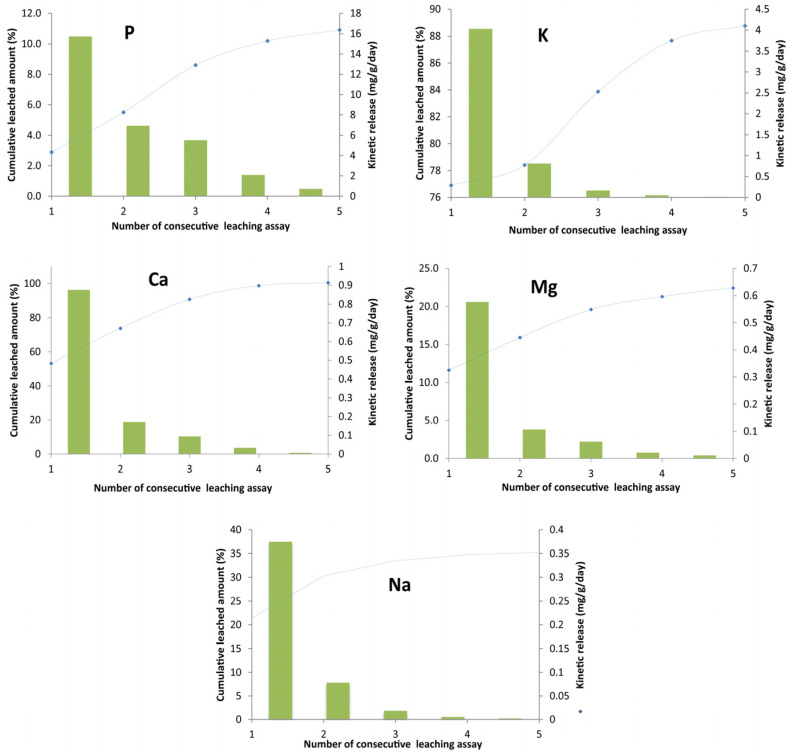
The effects of successive leaching assays on the release of Na, K, Mg, Ca, and P from FS_treated_.

**Figure 11 molecules-28-01425-f011:**
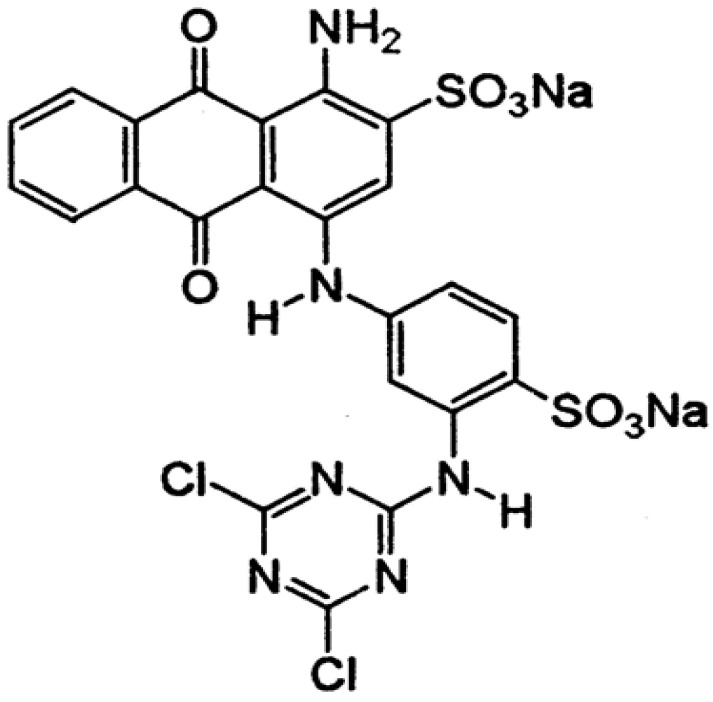
Structure of the RB4 dye.

**Table 1 molecules-28-01425-t001:** Results of EDS analysis for BC, nZVI-BC, and FS_treated_.

Element, %	BC	nZVI-BC	FS_treated_
C	37.43	27.61	19.84
O	28.83	22.17	31.40
Si	9.05	8.27	7.25
Ca	6.83	2.88	2.63
P	5.79	2.76	3.85
Fe	3.83	24.97	22.69
Al	3.33	3.17	2.94
K	2.89	3.75	3.62
Mg	1.27	1.37	1.34
Na	0.75	1.14	1.15
Cl	-	1.90	3.25

**Table 2 molecules-28-01425-t002:** Matrix of the experiment with the achieved decolorization efficiencies.

Sample	c(H_2_O_2_) (mM)	c(nZVI-BC) (mg/L)	c(RB4) (mg/L)	Removal Efficiency (%)
1	5.5	200	150	75.77
2	5.5	5	50	18.28
3	10	102.5	50	64.64
4	1	102.5	150	39.37
5	10	5	100	21.62
6	1	200	100	73.97
7	10	200	50	90.05
8	1	5	150	14.77
9	10	200	150	85.35
10	1	5	50	15.78
11	10	5	150	20.47
12	1	200	50	77.81
13	5.5	102.5	100	52.58
14	5.5	200	150	77.9
15	5.5	5	50	14.62
16	10	102.5	50	66.12
17	1	102.5	150	39.74
18	10	5	100	17.05
19	1	200	100	72.01
20	10	200	50	92.96
21	1	5	150	6.00
22	10	200	150	88.06
23	1	5	50	5.03
24	10	5	150	19.35
25	1	200	50	69.86
26	5.5	102.5	100	50.97
27	5.5	102.5	100	47.91
28	5.5	102.5	100	53.45

**Table 3 molecules-28-01425-t003:** The selected regression model (all interactions included).

Descriptive Factor	Value
R^2^	0.986522
R^2^ _adj_	0.982671
AIC	170.1597
BIC	173.2384
RMSE (Root Mean Square Error)	3.828282

**Table 4 molecules-28-01425-t004:** The Results of the physico-chemical characterization of the effluent before and after treatment.

Parameter	Before Treatment	After Treatment	Mineralization, %
pH	6.4	3.4	-
Conductivity (μS/cm)	72	280.6	-
BOD (mgO_2_/L)	0	16	-
COD (mgO_2_/L)	280	105	62.5%
TOC (mgC/L)	16	10.15	36.6%

**Table 5 molecules-28-01425-t005:** Compounds identified by GC/MS analysis after the applied Fenton treatment.

Compounds	Retention Time	Structure
Dibutyl phtalate	18.0703	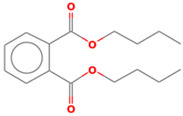
Phthalic acid isobutyl octyl ester	18.066	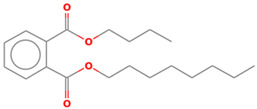
1,2-Benzenedicarboxylic acid bis (2-methyl propyl) ester	18.066	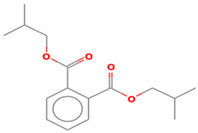

**Table 7 molecules-28-01425-t007:** Nutrient release efficiency after 1 and 5 leaching cycles from FS_treated_.

Element	Main Function of Nutrient for Plant Growth [49]	Released Amount at the First Leaching Assay (mg/g)	Percentage from the Contained Fraction in the Biochar (%)	Cumulative Released Amount at the Fifth Leaching Assay (mg/g)	Percentage from the Contained Fraction in the Biochar (%)
K	Primary—Resistance to diseases, fruit ripening, overall development, water regulation	4.03	76.9	6.40	89
P	Primary—Healthy vegetation growth, chlorophyll, formation of cells, basis for amino acids and proteins.	15.7	2.9	58.2	10.9
Ca	Secondary—Plant structure, growth and strength, resistance to diseases	0.87	53.1	1.66	100
Mg	Secondary—Vegetation development and growth, sugar formation, chlorophyll, fat formation, metabolism	0.58	11.6	1.12	22.4
Na	Micro- or trace nutrient—Plant growth, leaf color, formation of starch, enzyme development, and activity	0.38	21.4	0.62	35

**Table 8 molecules-28-01425-t008:** The experimental levels of the three process parameters.

Variable	Unit	Encoded Value	Level		
			−1	0	+1
c (H_2_O_2_)	mM	X_1_	1	5.5	10
c (nZVI-BC)	mg/L	X_3_	5	102.5	200
c (RB4)	mg/L	X_2_	50	100	150

## Data Availability

The datasets generated during and/or analyzed during the current study are not publicly available due to intellectual property or confidentiality concerns but are available from the corresponding author on reasonable request.

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
