# Peer review of "“Green” nZVI-Biochar as Fenton Catalyst: Perspective of Closing-the-Loop in Wastewater Treatment"

_molecules, 2023, doi:10.3390/molecules28031425_

Round 1

Reviewer 1 Report

General comments:

Manuscript ID (molecules-2129547). Titled "Green synthetized nZVI supported on sewage sludge based biochar for organic pollutant removal: Perspective of closing-the-loop in wastewater treatment." for possible publication in Molecules.

This paper’s topic is interesting but authors have to proofread the manuscript and improve the English and remove grammar mistakes. The referencing inside the manuscript needs to be corrected in many places and authors must add new references. The abstract should be improved by adding current research gap on this topic. The introduction section also needs improvements by adding new work already published globally. Moreover conclusion parts should be revised by giving a brief summary of your results and adding possible suggestions for future researchers.

Specific comments:

1.      Title is not attractive, clear and meaningful. Title should be revised by adding suitable words. Correct clos-ing-the-loop.

2.      Abstract should be revised by adding some words about research gap.

3.      Also add quantitative data in abstract. Avoid irrelevant discussion in abstract.

4.      Write objectives of your research paper clearly.

5.       Add some new work published in last 10 years. Delete old work and old references.

6.      Line 42: Introduction; please add global data about production of sewage sludge in world and also add previous work done related to this study in different countries of world.

7.      Line 80-81: give short details.

8.      Line 93,94, 96 and……: Correct in format and also remove this error in whole MS. (large-scale, cost-effectiveness).

9.      Fig 1. Write correct spelling of Pirolysis. Remove spelling mistakes in whole MS.

10.  Fig 3, 7 and 10. Make clear figures.

11.  Line 221: Chapter 3.6? correct it

12.  Line 223: 28 experiments? Write suitable words.

13.  Line 226: correct paragraph setting. Revise in whole MS.

14.  Line 292: Masking?

15.  Line 327: Correct Grammar.

16.  Line 499: secondary and micro? Give correct details.

17.  Line 528: Fig 17? Correct it

18.  Line 572-574: Remove these.

19.  Line 590-591: give GPS points and details of location.

20.  Also add research gap in introduction section.

21.  Add these new works published in discussion part,  DOI:https://doi.org/10.1016/j.chemosphere.2019.124679

22.  Authors must add details about mechanisms involved in the sections.

23.  Please include following details in Conclusion section.

·         Write brief summary of your findings with solid justification.

·         Add some words about novelty of your research work.

·         Give your suggestions for future research work about this topic.

Author Response

Dear Reviewer 1, 

please find enclosed the responses to you comments in the attachment. 

Best regards.

Reviewer 2 Report

The manuscript " Green synthetized nZVI supported on sewage sludge based  biochar for organic pollutant removal: Perspective of closing-the-loop in wastewater treatment " is of practical importance and will be of interest to readers of Molecules.

After reading this manuscript, I had a few questions and comments.

1.     In Table 4 the unit of measure for Conductivity is μS/c. What does "c" mean?

2.     Line 157. Reference is made to figure 3b. Is this not an error? I think the reference should be to figure 3a.

3.     Figure 6. It is necessary to explain the figure. It is not clear from the text of the manuscript what kind of samples were discoloured. Are these different samples or 5 replicates of the same water sample with a RB4 concentration of 50 mg/l?

4.     Figure 10. In the Materials and Methods section it is said that the leaching rate of the elements was calculated as Cn/D.  Cn is the concentration in the equilibrium liquid phase after two days of agitation on the rotator and D is respectively 2. If it is so, why on the abscissa axis of figure 10 are days? In my opinion the abscissa should reflect the number of consecutive treatments (leaching procedures).  For these reasons, the results of sequential leaching cannot be interpreted as "leaching kinetics". The kinetic curve is the dependence of concentration on time. In Figure 10 the authors have presented the average leaching rate for each of the five sequential leaching procedures. 

5.     Figures 3, 7 and 10 are of poor quality and difficult to understand.  

The overall conclusion is that the authors have obtained interesting experimental material and proposed a new approach to wastewater treatment. I have no doubt that the research carried out will be of interest to the readers of the journal. I recommend this manuscript for publication after minor corrections and clarifications.

Author Response

Dear Reviewer 2, 

please find the enclosed responses to you comments in the attachment. 

Best regards.
